# When does Observational Data Teach Latent Dynamics? Understanding Control Misalignment with Synthetic Tasks

## Abstract

Deep generative models are increasingly embedded in applications in robotics, simulation, and image/video/audio synthesis. In these settings, data likelihoods may depend on hidden "control parameters" not directly observed at training time (e.g., speed, energy, transition rule). Although standard loss functions do not enforce distributional alignment over such hidden variables, practitioners often assume that models generate samples with controls reflecting those priors. We identify *Control Misalignment* (CM): model generations consistently violate distributional alignment with the control prior in patterned ways across tasks and architectures, posing significant safety and fairness concerns. We first catalogue the prevalence of CM in real-world vignettes: distribution shifts of movement speed in D4RL motion planning, total energy in double-pendulum physical simulation, and speaking rate in Tacotron2 speech synthesis. Next, we probe *when*, and *why*, such drifts emerge, testing confounds against carefully constructed synthetic tasks with known controls and tunable chaoticity. Through this, we characterize the mechanism behind CM: error signatures in data space are transported through ill-conditioned or ambiguous recovery procedures into coherent control-space malapportionment. We verify this mechanistic interpretation by constructing a minimal toy system that reproduces the defining characteristics of CM. Finally, we apply our learnings to propose mitigations, analyzing which ones are theoretically sound or empirically effective versus not. Overall, our work provides a precise mechanistic account as to why, and under what conditions, Control Misalignment arises in generative pipelines.

## 1 Introduction

Deep learning is now widely used to train neural networks taking on various shapes, sizes, and applications. For example, they are trusted as surrogates for complex real-world processes, from robotic motion planning (Janner et al., 2022) and dynamical simulation (Li et al., 2023) to speech synthesis (Shen et al., 2018; Kong et al., 2020). In these scientific and engineering domains, the observable data $x$ is often governed by semantically meaningful *control parameters*: examples include the speed of a trajectory, rule of a physical system, or the speaking rate of an utterance. When training is unconditional, these controls remain latent; the model sees only the marginal data distribution $q(x) = \int p(x \mid r)\pi(r)dr$. Standard generative objectives (e.g., NLL, ELBO, Score Matching) incentivize the model to match this data marginal $q(x)$. Critically, however, approximating the data marginal does not guarantee that the implicit distribution over latent controls is preserved. We identify this phenomenon as *Control Misalignment* (CM), encapsulating how the distribution of controls recovered from generated samples systematically deviates from the training prior $\pi(r)$, even when the generative model appears to achieve high fidelity on standard metrics.

*Why does this occur?* We argue for a mechanistic explanation based on density transport through control recovery: structured observation-space errors can be low-cost under marginal objectives yet become large recovered-control shifts when recovery is sensitive or ambiguous. We test this perspective by moving from real-world vignettes to controlled synthetic environments that isolate sensitivity (tent/logistic/sinusoid) and ambiguity (folded map), and by validating the mechanism in a minimal toy system. Our contributions are as follows:

- **Characterizing CM.** We show CM in Maze2D speed, pendulum energy, and Tacotron2 speaking rate (Sec. 2.1). We then formalize Control Misalignment as divergence between recovered-control pullbacks and the intended prior (Def. 1).
- **Mechanism + validation.** We explain CM as density transport through recovery and validate it on sensitivity/ambiguity benchmarks (Sec. 2).
- **Mitigations.** We analyze iterative prior reweighting and conditioning, showing limits under ambiguity and gains with control labels (Sec. 4).

**Prior work.** Many works document systematic sampling biases in diffusion and autoregressive models—mode interpolation, signal leak, spectral and exposure bias—and show that aggregate metrics can hide miscalibration (Aithal et al., 2024; Everaert et al., 2024; Kadkhodaie et al., 2023; Schmidt, 2019; Meister & Cotterell, 2021). CM also connects to inverse problem theory, where sensitivity and ambiguity turn small observation errors into large parameter errors (Tikhonov & Arsenin, 1977; Tarantola, 2005; Bellman & Åström, 1970).

## 2 EVIDENCE AND MECHANISM OF CM

### 2.1 Control Misalignment in Deployed Pipelines.
We first examine three deployed pipelines and consistently observe shifts in semantically meaningful latent controls. We report numerical summaries in Appx. I.

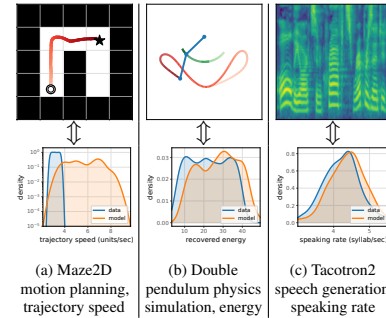

(a) Maze2D motion planning, trajectory speed

(b) Double pendulum physics simulation, energy

(c) Tacotron2 speech generation, speaking rate

Figure 1: Perceptually reasonable generation can be significantly misaligned in control space.

- **Maze2D diffusion planning.** We use a pretrained Diffuser-style trajectory model to generate goal-conditioned plans in D4RL Maze2D U-maze (Fu et al., 2020; Janner et al., 2022), recover trajectory speed, and resample to enforce a uniform speed prior; a model trained on this set shifts toward higher speeds (Fig. 1a).
- **Physical simulation.** We simulate an energy-conserving double pendulum with initial energy $E$ drawn uniformly, train an unconditional diffusion model on trajectories, and recover energy by finite-differencing positions and evaluating the energy functional. While the model's samples look plausible, their recovered energy distribution deviates from the intended uniform prior, shifting towards higher energies (Fig. 1b).
- **Tacotron2 speech synthesis.** We evaluate Tacotron2 (Shen et al., 2018) on LJ Speech (Ito & Johnson, 2017), recover speaking rate using wav2vec 2.0–based transcription (Baevski et al., 2020) and syllable counts with a transcript-consistency filter. Appx. I.3 details the filtering and syllable-counting procedure. The recovered speaking-rate marginal for generations trends faster than that of the training data (Fig. 1c).

### 2.2 Density Transport Through Recovery.
The vignettes above share a common structure: the data are indexed by latent controls, but the model is trained unconditionally on the marginal over observations. Let $r \in \mathcal{R} \subseteq \mathbb{R}$ denote one such control with intended prior $\pi(r)$, and let $x$ denote an observation. The data marginal is $q(x) = \int p(x \mid r)\pi(r)\,dr$, and an unconditional generative model learns $q_\theta(x)$. When a control can be recovered from observations, a recovery procedure maps each observation $x$ to a distribution $\rho(r \mid x)$ over controls; for a source distribution $S$ over observations, the induced recovered control marginal (pullback) is $\bar{\pi}_S(r) = \mathbb{E}_{x\sim S}[\rho(r \mid x)]$ (Eq. 1).

**Definition 1** (Control Misalignment). Let $\bar{\pi}_{\text{data}} := \bar{\pi}_q$ and $\bar{\pi}_{\text{model}} := \bar{\pi}_{q_\theta}$ under a shared recovery procedure. We quantify CM of a model $q_\theta$ with respect to an intended prior $\pi$ by the total-variation divergence $\text{TV}(\bar{\pi}_{q_\theta}, \pi) = \frac{1}{2}\int_{\mathbb{R}} |\bar{\pi}_{q_\theta}(r) - \pi(r)|\,dr$. We say a model exhibits CM when this divergence substantially exceeds the data baseline, i.e., when $\text{TV}(\bar{\pi}_{\text{model}}, \pi) \gg \text{TV}(\bar{\pi}_{\text{data}}, \pi)$.
Even when recovery is certified on data (so $\bar{\pi}_q \approx \pi$), applying the same map to model samples can yield a different pullback. We unpack the mechanism in three steps: when CM can be large, what determines the misalignment signature, and why recovery is high-gain.

- **When can CM be large?** If $\rho$ is $L$-Lipschitz in total variation, then $\text{TV}(\bar{\pi}_S, \bar{\pi}_T) \leq L\,W_1(S, T)$ (Eq. 5). The triangle inequality gives $\text{TV}(\bar{\pi}_{q_\theta}, \pi) \leq L\,W_1(q_\theta, q) + \text{TV}(\bar{\pi}_q, \pi)$, so large CM requires either a sizeable observation-space discrepancy or a high recovery gain.
- **What determines the misalignment signature?** Writing $\Delta_\theta(x) = q_\theta(x) - q(x)$ and $\tau_\theta(r) = \bar{\pi}_{q_\theta}(r) - \bar{\pi}_q(r)$, the transport identity $\tau_\theta(r) = \int \Delta_\theta(x)\,\rho(r \mid x)\,dx$ (Eq. 2) shows that signed model error is redistributed across controls by the recovery map, so low-cost deviations in $x$ that align with recovery gain explain the coherent drift patterns in the vignettes.

- **What makes recovery high-gain?** Sensitivity means $\rho(\cdot \mid x)$ changes rapidly with $x$ (large $L$), while ambiguity means multiple controls assign comparable mass to the same $x$, so small perturbations swap mass between competing explanations.

### 2.3 When and Where CM Appears.
We now instantiate sensitivity and ambiguity in minimal synthetic benchmarks where the data generation process and recovery procedure are fully specified.

#### 2.3.1 Dynamical sensitivity.
We begin with dynamical systems where the forward map becomes increasingly sensitive as a function of the control. The tent map provides a minimal one-dimensional setting: $x_{t+1} = f_r(x_t) = r \cdot \min\{x_t, 1-x_t\}$ with $r \in [0, 2]$ and $x_0 = 0.25$. Appx. D.2 defines the logistic and sinusoid comparison families. Unless stated otherwise, we use a uniform prior over $r$, train an unconditional diffusion model (Ho et al., 2020; Song et al., 2021) on the marginal over trajectories, and recover controls using the shared procedure in Eq. 3. The recovered control pullback on ground-truth data stays close to the baseline. By contrast, the unconditional model exhibits substantial misalignment, with the largest deviations in the chaotic regime. In the logistic-map comparison, misalignment likewise concentrates in sensitive regimes, while for the sinusoid family it is negligible; its controls remain well-separated in the MDS geometry under the same recovery interface. Quantitatively, in the baseline tent-map run, the model's recovered-control discrepancy is $\text{TV}(\bar{\pi}_{q_\theta}, \pi) \approx 0.13$ while the data baseline is $\text{TV}(\bar{\pi}_q, \pi) \approx 0.004$. Appx. F.1 and Appx. F.2 run ablations across estimators, model families, and horizons and show the qualitative pattern persists. We also supplement this understanding of sensitivity via Lyapunov exponents (rate of local trajectory separation) and MDS embeddings (where distinct controls collapse in observation space), defined in Appx. B. We show both in Fig. 2—the visualizations reveal that CM concentrates where the control-data relationship is particularly sensitive.

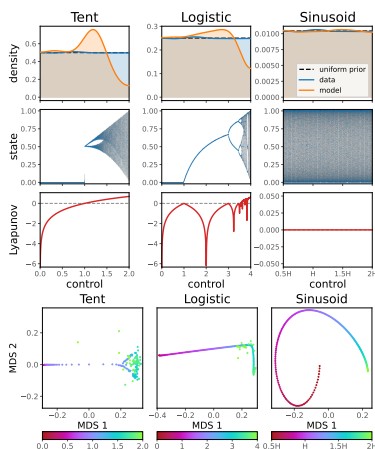

Figure 2: CM concentrates in regimes with chaotic and folded/clumped control, i.e., areas with high sensitivity.

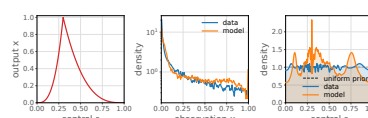

Figure 3: An asymmetric many-to-one simulator makes the inverse ambiguous, inducing CM.

#### 2.3.2 Recovery ambiguity.
To isolate ambiguity in the recovery interface, we construct a minimal one-dimensional inverse problem where $r$ is intrinsically non-identifiable from $x$. Appx. H details the folded-map construction. Let $r \sim \text{Unif}[0, 1]$ and define a deterministic simulator $g : [0, 1] \to [0, 1]$ by an asymmetric fold at $c \in (0, 1)$, and observe $x = g(r) + \epsilon$ with $\epsilon \sim \mathcal{N}(0, \sigma_{\min}^2)$. For most $x$, there are two distinct controls $r \neq r'$ with $g(r) = g(r')$, so any recovery based only on $x$ is necessarily ambiguous. Even though a single posterior $\rho(r \mid x)$ can have two sharp modes, the pullback on ground-truth data remains near the intended prior because recovery matches the observation model; ambiguity alone does not create CM. CM appears when a model perturbs the observation distribution, and recovery behaves like a soft competition between preimage branches, so small biases in where $q_\theta$ concentrates probability can change which branch "wins" (Fig. 3).

Overall, these tasks support the mechanism from Sec. 2: recovery-side sensitivity and ambiguity localize where CM can be large, and the signature of misalignment depends on the interaction between the control-data map and the model's preferred observation-space errors.

## 3 EXPLAINING THE VIGNETTES

With the mechanism and the synthetic benchmarks from Sec. 2 in hand, we can interpret each deployed vignette as *density transport through recovery*: the generator can make small, plausible observation-space deviations that become large recovered-control shifts when they align with high-gain (sensitive) or ambiguous directions of the recovery interface.

- **Maze2D speed misalignment.** In Maze2D, the recovered control is a simple functional of the plan, $v(x) = \frac{1}{(H-1)\Delta t} \sum_t \|q_{t+1} - q_t\|_2$ (Appx. I.1). In Fig. 4a, we add feasible normal-direction perturbations scaled to a fixed trajectory MSE budget and vary only their roughness in time; increasingly jittery perturbations induce much larger shifts in recovered speed, illustrating how low-cost deviations can yield large control shifts. In this vignette, the resulting recovered-speed misalignment is large (TV $\approx 0.92$; Appx. I).

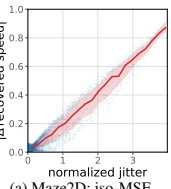

(a) Maze2D: iso-MSE jitter vs. speed

- **Pendulum energy misalignment.** For the double pendulum, we recover energy from position sequences by finite differences and the standard energy functional (Appx. I.2). This recovery is sensitive (differentiation) and ambiguous off the simulator manifold: model samples drift in recovered energy over time, so there is no single energy value that an energy-conserving simulator could have generated. We visualize this in Fig. 4b, where a representative model trajectory violates conservation in a way that forces recovery to pick a single energy anyway; the recovered-energy discrepancy is TV $\approx 0.12$ (Appx. I).

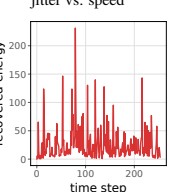

(b) Pendulum: energy conservation violated

- **Tacotron2 speaking-rate misalignment.** Speaking rate is estimated as syllables-per-second after ASR-based filtering; Sec. 2.1 and Appx. I.3 detail the pipeline. This recovery is sensitive to time-axis distortions: time warps can change syllable rate without requiring large spectral changes. We construct iso-loss time-warp pairs for each utterance and find a consistent asymmetry: equal mel loss often permits substantially larger speed-ups than slow-downs (Fig. 4c), which provides a natural explanation for why an unconditional model can misalign toward faster recovered speaking rates (TV $\approx 0.13$; Appx. I).

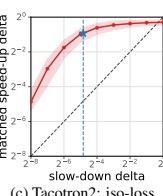

(c) Tacotron2: iso-loss time-warp asymmetry

Figure 4

## 4 MITIGATIONS

Approximating the observation marginal can still admit small deviations that recovery amplifies into CM; so, mitigations must reduce such deviations, reshape the training mixture over controls, or make the relevant variation explicit

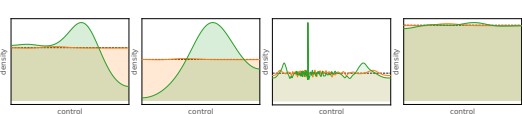

More capacity | Autoregressive | Prior correction | Conditioning

Figure 5: Mitigations; most fail to fully fix CM.

during training. We explore several approaches below (accompanied by Fig. 5 and Appx. F).

- **Improving fidelity and inductive biases.** Because Eq. 5 links observation-space error to pullback error, higher fidelity and inductive biases that suppress recovery-amplified artifacts are beneficial, but indirect; on the tent map, neither a wider model nor an autoregressive generator reduce CM.
- **Iterative prior correction (IPC).** When we can sample data at chosen controls, IPC reweights the training mixture via $\pi_{\text{train}}^{(k+1)}(r) \propto \pi(r)/\left(\bar{\pi}_{q_{\theta(k)}}(r) + \varepsilon\right)$ and retrains. The folded-map benchmark shows IPC improves alignment but does not eliminate it, since ambiguity continues to distribute posterior mass across competing preimages.
- **Conditional modeling.** If the control (or a reliable proxy) is known before training, we can train $p_\theta(x \mid r)$ and sample with $r \sim \pi$, making the variation explicit; conditioning largely restores alignment. In low-gain (near-unambiguous) settings such as the sinusoid family, unconditional models can already sit near the data baseline, so conditioning may offer little utility.

## 5 CONCLUSION

We identify CM, an omitted-variable failure mode of unconditional generative models: even when recovery validates on data, applying it to generations can expose shifts in the implied control mixture, so approximating $q(x)$ need not cover unobserved controls. Across deployed pipelines and synthetic benchmarks, we find a common mechanism: unconditional training makes certain structured deviations in observation space cheap, and sensitive or ambiguous recovery interfaces transport those deviations into coherent misalignment in recovered controls. Our mitigation results also suggest a practical lesson: when a salient control can be identified (or approximated by a reliable proxy), conditioning on it can make the model respect alignment while control-blind strategies cannot. More broadly, this motivates treating control discovery and measurement as a key focus of model development: without explicit controls to condition on or probe via recovery, observation-space fidelity (and capacity scaling) can mask systematic distortions and biases along salient control axes.

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

## A  ADDITIONAL RELATED WORK

A growing body of work reveals systematic sampling biases across model families. For example, diffusion models place excess probability mass along low-density bridges between modes (Aithal et al., 2024), exhibit signal-leak artifacts from prior mismatch (Everaert et al., 2024), and preferentially learn smooth structure due to spectral bias (Wang & Pehlevan, 2025; Kadkhodaie et al., 2023). Recent work also studies bias and its amplification in diffusion models, and proposes guidance mechanisms to improve coverage of underrepresented groups (Um et al., 2023; Hakemi et al., 2025; Roos et al., 2025). Autoregressive models suffer from exposure bias—a mismatch between teacher-forced training on ground-truth sequences and free-running inference on the model's own predictions, causing errors to compound quadratically with sequence length (Schmidt, 2019). These biases often evade detection because standard metrics such as likelihood, inception score, perplexity, and FID measure aggregate fidelity rather than calibration across the full support of a latent variable. A model can therefore appear well-trained while harboring systematic distortions (Naeem et al., 2020; Meister & Cotterell, 2021). In some sense, these data-space artifacts can be viewed as each architecture's preferred method of spending an error budget under a marginal objective.

Separately, the premise of CM—involving a hidden variable that dictates the data generation process—is rooted in classical inverse problem theory (Tikhonov & Arsenin, 1977; Tarantola, 2005; Kaipio & Somersalo, 2005). When an inverse map is sensitive or ambiguous, small observation-space errors propagate into large parameter-space errors. This instability is well-studied in dynamical systems parameter estimation (Bellman & Åström, 1970), where structural identifiability questions arise even with noiseless data. The crucial difference here is that the "perturbation" is not random noise, but the structured bias of a learned distribution of deep generative models, visible only through inversion.

# B   SUPPLEMENTARY DEFINITIONS

This appendix records definitions and the shared recovery interface used throughout the main text and appendices.

Throughout, a recovery procedure maps each observation $x$ to a distribution $\rho(r \mid x)$ over controls. Recovery can be probabilistic (posterior-like) or deterministic, in which case $\rho(r \mid x)$ is a point mass at a recovered control $\hat{r}(x)$.

For any source distribution $S$ over observations, the recovered control marginal (pullback) is

$$\bar{\pi}_S(r) = \mathbb{E}_{x \sim S}[\rho(r \mid x)]. \tag{1}$$

Writing $\Delta_\theta(x) = q_\theta(x) - q(x)$ and $\tau_\theta(r) = \bar{\pi}_{q_\theta}(r) - \bar{\pi}_q(r)$, the density-transport identity is

$$\tau_\theta(r) = \int \Delta_\theta(x)\, \rho(r \mid x)\, dx. \tag{2}$$

In most synthetic benchmarks, a forward simulator $g : \mathcal{R} \to \mathcal{X}$ is available and recovery is implemented by comparing observations to simulator references. In principle, one could operationalize recovery by Bayes' rule, $\rho(r \mid x) \propto p(x \mid r)\pi(r)$, but in many settings the forward map is (approximately) deterministic: $x = g(r)$. Without an explicit noise model, the corresponding likelihood is degenerate, $p(x \mid r) = \delta(x - g(r))$, so the posterior collapses to exact (or nearest-neighbor) matching and becomes brittle to discretization and to the fact that model samples are not exact simulator outputs. We therefore adopt an observation model that treats deviations from the simulator as noise, $x = g(r) + \varepsilon$, and use the resulting likelihood to define a stable, shared recovery procedure across data and model samples. Concretely, on a fixed control grid $\{r_j\}_{j=1}^J$ we compare $x$ to simulated references $g(r_j)$ and convert discrepancies into an adaptive Gaussian likelihood:

$$e_j(x) = \|x - g(r_j)\|_2^2,$$
$$\rho(r_j \mid x) \propto \exp\left(-\frac{e_j(x)}{2\sigma^2(x)}\right) \pi(r_j),$$
$$\sigma^2(x) = \max\left(\min_j e_j(x), \sigma_{\min}^2\right). \tag{3}$$

Here $\sigma_{\min} > 0$ is a numerical floor and $g(r)$ denotes the simulator output used as a recovery reference (e.g., a fixed initial-condition rollout for dynamical systems, or a point on a geometric curve). Sec. D gives implementation details for the deployed vignettes and synthetic benchmarks.

We also record the stability assumption used in the mechanism section and the resulting bound:

$$\mathrm{TV}(\rho(\cdot \mid x), \rho(\cdot \mid x')) \leq L \|x - x'\|_2, \tag{4}$$
$$\mathrm{TV}(\bar{\pi}_S, \bar{\pi}_T) \leq L\, W_1(S, T). \tag{5}$$

**Lyapunov exponent (sensitivity).**  The (maximal) Lyapunov exponent measures the exponential rate at which nearby trajectories separate under a dynamical system. Positive exponents indicate sensitivity to initial conditions; in the tent-map experiments, we estimate it to localize high-gain regimes where small control changes lead to large observation changes.

**Multidimensional scaling (MDS).**  MDS (Torgerson, 1952) embeds points into a low-dimensional Euclidean space while approximately preserving pairwise distances from a given distance matrix. We use it to visualize the control-geometry induced by observation-space distances, revealing where distinct controls become close in $x$-space.

## C  TEMPLATE FOR MECHANISTIC ANALYSIS

Real pipelines mix together many effects such as learning dynamics, measurement noise, recovery heuristics, and inference-time procedures. To isolate the mechanism, we build synthetic benchmarks that keep the structure of interest—a control-indexed data family together with a recovery procedure—while stripping away most confounds. Concretely, the benchmark makes three ingredients explicit: an intended prior $\pi(r)$, a simulator that generates $x \sim p(x \mid r)$, and a fixed recovery procedure that can be validated on ground-truth data. We then apply the *same* recovery procedure to both data and model samples. Under this shared interface, differences in recovered-control pullbacks reflect the generator, recovery procedure, and the prior over $r$.

The evaluation template, depicted in Fig. 6, is:

1. Sample a control $r \sim \pi(r)$ and generate $x \sim p(x \mid r)$.
2. Train an unconditional model $q_\theta(x) \approx q(x) = \int p(x \mid r)\pi(r)\,dr$.
3. Recover a distribution $\rho(r \mid x)$ under the shared interface in Eq. 3.
4. Compute pullbacks $\bar{\pi}_q$ and $\bar{\pi}_{q_\theta}$ via Eq. 1 and report CM via Def. 1.

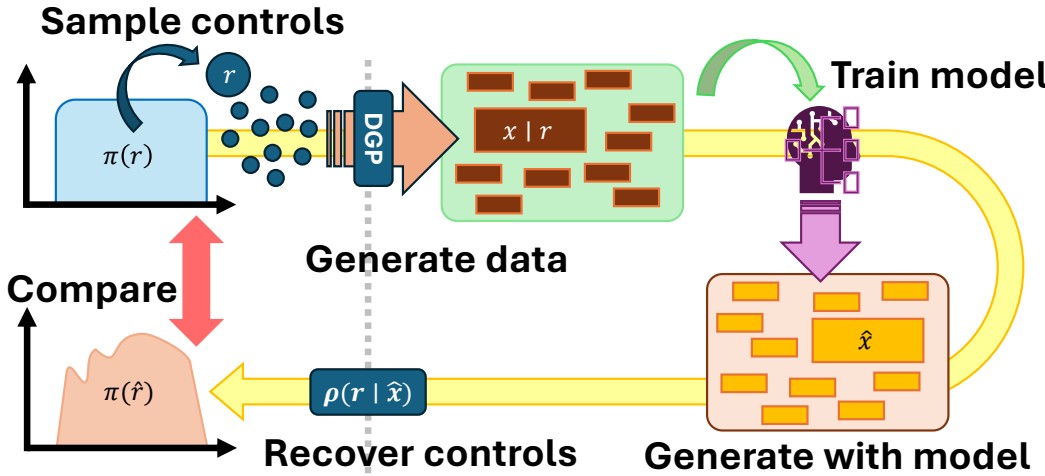

Figure 6: **Synthetic framework to study the mechanism of CM.** The steps are: sample $r$ values from a prior distribution; use the data-generation process (DGP) to obtain observations following those $r$ values; train a model only on observations; generate samples with the model; recover what controls correspond to each generated sample; and compare the implied control distribution to the intended prior. This template admits modularity of parts like the prior, simulator, model, and recovery procedure.

Note that for most experiments, we prevent the shape of the prior from becoming a separate confound by using a uniform prior over control parameters. In principle, this can be swapped for other priors; however, it is not necessary to test this directly, as such pipelines can be reparameterized by pushing the distribution into the generator definition to define a new generator with a uniform control prior. Concretely, if there is a transport map $T : [0,1]^d \to \mathcal{R}$ such that $T_\# \mathrm{Unif}([0,1]^d) = \pi$, then writing $r = T(u)$ with $u \sim \mathrm{Unif}([0,1]^d)$ yields the same marginal

$$q(x) = \int p(x \mid r)\,\pi(r)\,dr = \int_{[0,1]^d} p(x \mid T(u))\,du. \tag{6}$$

In the scalar case, $T(u) = F_\pi^{-1}(u)$ is the usual CDF transform.

# D  SETUP AND SHARED PROTOCOLS

## D.1  MODEL ARCHITECTURES AND TRAINING DETAILS

**Overview.**  This section documents the generator architectures and training hyperparameters used for results reported in the main text and appendix ablations. When multiple variants appear (e.g., a wider denoiser, a conditional model, or an autoregressive baseline), we list the concrete parameter choices in the tent-map hypothesis ladder (Sec. F) and in the deployed vignettes (Sec. I).

**Primary synthetic generator: JannerUNet1d diffusion.**  For the tent/logistic/sinusoid dynamical families (and for most diffusion variants in the hypothesis ladder), we use a 1D U-Net denoiser adapted from the Diffuser-style architecture of Janner et al. (2022) via a lightweight fork of an open-source diffusion implementation. The denoiser takes a length-$H$ trajectory and predicts the diffusion noise (or $x_0$ for the objective ablation), and diffusion sampling is performed with either DDPM or DDIM. Unless stated otherwise, the synthetic diffusion setup is:

- **Data representation:** trajectories $x_{0:H} \in [0, 1]^H$ are linearly mapped to $[-1, 1]$ for diffusion training; samples are mapped back to $[0, 1]$ for recovery and plotting.
- **Architecture (denoiser):** a Diffuser-style 1D U-Net with **input channels** 1, **base width** 32, **channel multipliers** $(1, 2, 2, 2)$, **kernel size** 3, **timestep embedding dim** 32 (positional), **activation** Mish, and **normalization** (groupnorm-style) throughout; no attention blocks are enabled in the baseline.
- **Optimizer:** AdamW with learning rate $3 \cdot 10^{-4}$, weight decay $10^{-5}$, batch size 1024.
- **Training length:** 20 epochs in the full tent-map preset (and scaled proportionally in shorter presets used for quick runs).
- **Sampling:** DDPM with 256 sampling steps (unless an ablation specifies DDIM with fewer steps).

**Control-conditioned diffusion (conditioning mitigation).**  For the control-conditioning mitigation, we keep the same diffusion backbone but inject a control embedding into the denoiser:

- **Control embedding:** $r$ is mapped to $u \in [0, 1]$ using the known control range and encoded with a sinusoidal feature map of dimension 128.
- **Injection:** FiLM-style scale/shift modulation in residual blocks and feature-channel concatenation of a projected control embedding (concatenation width 64).
- **Backbone changes:** base width 64 (wider U-Net), trained for 60 epochs; sampling uses DDIM.

**Alternative generator families (hypothesis ladder baselines).**  To test architecture dependence under the same recovery interface, we also report results for:

- **MLP diffusion denoiser:** a time-input MLP denoiser with hidden widths $(1088, 1088, 1088)$ (GELU activations) trained with the same denoising objective but without convolutional locality. This backend uses a log-linear $\sigma$ schedule with $N = 256$ noise levels over $\sigma \in [5 \cdot 10^{-3}, 3]$.
- **Autoregressive transformer:** sequences are discretized into 1024 bins and modeled token-by-token with a causal Transformer with $d_{\mathrm{model}} = 216$, 4 layers, 6 heads, dropout 0.1, GELU MLP blocks, and tied input/output embeddings. We train with AdamW (learning rate $3 \cdot 10^{-4}$, weight decay $10^{-2}$) with early stopping on a 2% validation split (patience 2); sampling uses temperature 1.0 with top-$k$ truncation ($k = 64$).
- **VAE:** a simple MLP VAE on the flattened trajectory with latent dimension 1, hidden widths $(1088, 1088)$, and a $\beta$-VAE objective ($\beta = 1$ with KL warmup over 5 epochs). Samples are generated by drawing $z \sim \mathcal{N}(0, I)$ and decoding.

**Ambiguity benchmark generator (folded-map IPC).**  For the ambiguity benchmark in Sec. 2.3.2 and Sec. H, we use a deliberately minimal 1D diffusion generator trained on scalars $x \in [0, 1]$ produced by the folded forward map:

- **Denoiser:** an MLP $\epsilon_\theta(x_t, t)$ with hidden size 128 and 3 hidden layers (SiLU activations).
- **Diffusion:** $T = 100$ steps with a linear $\beta$ schedule from $10^{-4}$ to $2 \cdot 10^{-2}$; trained for 15,000 optimization steps with Adam (learning rate $2 \cdot 10^{-4}$) and EMA decay 0.999.

**Deployed vignettes: Maze2D and double pendulum.** For Maze2D planning, we use a goal-conditioned diffusion planner for inpainting and train a separate diffusion model from scratch on a speed-uniform resampled dataset:

- **Pretrained proposal model:** a publicly available pretrained Maze2D trajectory diffusion model, used only to generate an initial pool for speed-uniform resampling.
- **Uniform-speed scratch model (used for CPM evaluation):** a 1D U-Net diffusion model (block channels $(32, 128, 256)$; Mish; group normalization) trained with a $64$-step diffusion schedule to predict the denoised sample. Training uses batch size $4096$, learning rate $10^{-4}$, $25{,}000$ optimization steps, and EMA decay $0.999$.
- **Double pendulum diffusion:** a Diffuser-style 1D U-Net with input channels 2 (angles $(q_1, q_2)$), horizon 256, and the same baseline width/multipliers as above; trained for 50 epochs with batch size 64 and learning rate $10^{-3}$.

**Deployed vignette: Tacotron2.** For the speech vignette, we do not retrain the generator; we use a publicly available pretrained Tacotron2 model with a neural vocoder for synthesis, and we apply the speaking-rate recovery protocol described in Sec. 2.1 and Sec. I.3.

### D.2  ADDITIONAL DETAILS

**Synthetic data generation (defaults).** Unless stated otherwise (e.g., horizon-length or system-family ablations), we use:

- **Initial condition:** fixed $x_0 = 0.25$ for the dynamical families (tent/logistic/sinusoid), matching the main text.
- **Prior:** uniform $\pi(r)$ over the system's stated control range.
- **Numerical stabilization:** for chaotic iterated maps, we discretize intermediate states onto a fixed grid (1024 bins on $[0, 1]$) to limit floating-point accumulation in long rollouts; this discretization is applied equally wherever the simulator is used (data generation and simulator references for recovery).

We revisit how these defaults interact with dynamical sensitivity (e.g., high-chaos regimes) in Sec. F.

**Recovery interface (simulator-backed posterior).** When a forward simulator is available, recovery is implemented on a fixed control grid $\{r_j\}_{j=1}^J$ spanning $\mathcal{R}$. For each $r_j$ we precompute the simulator reference $g(r_j)$, and for a given observation $x$ we define the squared discrepancy $e_j(x) = \|x - g(r_j)\|_2^2$ (trajectory $\ell_2$, unless an estimator ablation specifies a different norm). We then form a discrete posterior over the grid using the adaptive Gaussian likelihood from the main text (Eq. 3), and we report both: (i) *point estimates* (posterior mode or posterior mean, depending on the ablation tag), and (ii) *mixture pullbacks* obtained by averaging posteriors over samples (used for the primary pullback plots and TV numbers). This same recovery interface is the shared measurement layer for the hypothesis ladder (Sec. F) and is the mechanism by which ambiguity can amplify CM in the folded-map benchmark (Sec. H). A stability view of this mapping (and why sensitivity/ambiguity create high-gain regimes) is given in Sec. E.

**Control misalignment metric.** We quantify CM by total variation between the recovered-control pullback and the intended prior, $\mathrm{TV}(\bar{\pi}_S, \pi)$. For grid-based recovery this is computed on the same discretization used to represent $\bar{\pi}_S$ on $\{r_j\}$, and we report both the data baseline and the model value for each experiment. For deployed vignettes where recovery uses proxies (e.g., finite differencing, ASR-based heuristics), we follow the same shared-recovery convention and report the same family of summary statistics; see Sec. I.

**Default synthetic generator and sampling.** Unless an ablation changes the model family, we use an unconditional diffusion generator (Sec. D.1) and sample with 256 DDPM steps; sampler ablations use DDIM with fewer steps. The concrete ablations (and the corresponding recovered-control pullbacks) are described in Sec. F.

# E  MECHANISM: PROOF DETAILS

This section provides proof details for the quantitative stability statement in Sec. 2. Let $\rho(\cdot \mid x)$ be a recovery distribution on $\mathcal{R}$ given an observation $x$, and for any source distribution $S$ over observations recall the pullback $\bar{\pi}_S = \int \rho(\cdot \mid x)\, S(dx)$.

**A general Wasserstein-1 to total-variation bound.**  Assume there exists a constant $L$ such that for all $x, x'$,
$$\mathrm{TV}(\rho(\cdot \mid x), \rho(\cdot \mid x')) \leq L\|x - x'\|_2,$$
as in Eq. 4. Equivalently, one may take $L := \sup_{x \neq x'} \mathrm{TV}(\rho(\cdot \mid x), \rho(\cdot \mid x'))/\|x - x'\|_2$ over the region of interest (so $L$ can be large when recovery is highly sensitive). Then for any two source distributions $S, T$ over observations,
$$\mathrm{TV}(\bar{\pi}_S, \bar{\pi}_T) \leq L\, W_1(S, T),$$
as in Eq. 5. *Proof.* Let $\gamma$ be any coupling of $S$ and $T$, with $(X, Y) \sim \gamma$. Using $\mathrm{TV}(\mu, \nu) = \frac{1}{2}\|\mu - \nu\|_1$ and convexity of $\|\cdot\|_1$,
$$\begin{aligned}
\mathrm{TV}(\bar{\pi}_S, \bar{\pi}_T) &= \mathrm{TV}(\mathbb{E}[\rho(\cdot \mid X)], \mathbb{E}[\rho(\cdot \mid Y)]) \\
&\leq \mathbb{E}\big[\mathrm{TV}(\rho(\cdot \mid X), \rho(\cdot \mid Y))\big] \\
&\leq L\, \mathbb{E}\|X - Y\|_2.
\end{aligned}$$
Taking the infimum over couplings $\gamma$ yields $W_1(S, T)$.

**Adaptive Gaussian recovery: sensitivity scaling.**  For the adaptive Gaussian recovery rule in Eq. 3, the map $x \mapsto \rho(\cdot \mid x)$ is locally TV-Lipschitz away from a set of measure-zero tie points where the nearest reference changes, with a bound scaling inversely with the local bandwidth $\sigma(x)$. This supports the main-text statement that recovery sensitivity increases as $\sigma(x)$ decreases.

**Ambiguity factor in two-way competitions.**  When recovery is dominated by two competing explanations with log-weights $\ell_a(x), \ell_b(x)$, define $p(x) = \exp(\ell_a(x))/(\exp(\ell_a(x)) + \exp(\ell_b(x)))$. Then
$$\nabla_x p(x) = p(x)(1 - p(x))\, \nabla_x(\ell_a(x) - \ell_b(x)),$$
so near-ties ($p \approx \frac{1}{2}$) are precisely where recovered mass swaps most readily between the two explanations.

# F  Synthetic Benchmarks: Ablations

We test whether the qualitative claims in the main text persist under changes to recovery estimation, training construction, optimization/sampling, and model family/inductive bias, while holding the recovery interface fixed.

## F.1  Numerical summary and pullbacks

**Baseline tent-map setting.**  Unless stated otherwise, the hypothesis ladder uses the tent map family with control range $r \in [0, 2]$ and horizon $H = 64$, with an intended uniform control prior $\pi(r)$. Each run trains on $n_{\text{train}} = 200{,}000$ trajectories and evaluates on $n_{\text{eval}} = 25{,}000$ trajectories under a shared recovery interface: a fixed control grid of size $J = 16{,}384$, an adaptive Gaussian likelihood with an $\ell_2$ discrepancy inside the exponent, and a posterior-mode point estimate used only for sample-level diagnostics. The baseline generator is an unconditional JannerUNet1d diffusion model with base width 32, kernel size 3, channel multipliers $(1, 2, 2, 2)$, trained for 20 epochs with batch size 1024, AdamW learning rate $3 \cdot 10^{-4}$ and EMA 0.999, and sampled with DDPM using 256 diffusion steps.

**Numerics (TV of recovered-control pullbacks).**  We report total variation between the recovered-control pullback and the intended prior for both the data baseline and the trained model. In the baseline tent-map run, $\text{TV}_{\text{data}} = 0.004$ while the unconditional model has $\text{TV}_{\text{model}} = 0.130$ for the posterior-mixture pullback (and $\text{TV}_{\text{model}} = 0.065$ under the point-estimator pullback).

## F.2  Investigating potential causes of CM on the tent map

| Hyp. | Experiment | System | $H$ | TV data | TV model (mix) | TV model (pt) |
|---|---|---|---|---|---|---|
| H0 | Baseline (tent map, diffusion) | tent | 64 | 0.004 | 0.130 | 0.065 |
| H1a | Estimator: $\ell_1$ discrepancy, posterior mode (point) | tent | 64 | 0.004 | 0.122 | 0.041 |
| H1b | Estimator: $\ell_2$ discrepancy, posterior mean (point) | tent | 64 | 0.004 | 0.128 | 0.163 |
| H1c | Estimator: $\ell_1$ discrepancy, posterior mean (point) | tent | 64 | 0.004 | 0.122 | 0.205 |
| H2 | Prior coverage: stratified control grid | tent | 64 | 0.000 | 0.126 | 0.059 |
| H3a | Optimization: short training | tent | 64 | 0.004 | 0.129 | 0.074 |
| H3b | Optimization: long training | tent | 64 | 0.004 | 0.122 | 0.048 |
| H4 | Capacity: wider denoiser | tent | 64 | 0.004 | 0.125 | 0.055 |
| H5 | Inductive bias: larger convolution kernel | tent | 64 | 0.004 | 0.127 | 0.061 |
| H6a | Model family: MLP denoiser | tent | 64 | 0.004 | 0.309 | 0.377 |
| H6b | Model family: autoregressive transformer | tent | 64 | 0.004 | 0.261 | 0.275 |
| H6c | Model family: VAE baseline | tent | 64 | 0.004 | 0.253 | 0.321 |
| H7 | Sampler: DDIM (fewer steps) | tent | 64 | 0.004 | 0.126 | 0.056 |
| H8 | Objective: $x_0$-prediction | tent | 64 | 0.004 | 0.126 | 0.060 |
| H9a | Context: shorter horizon | tent | 32 | 0.004 | 0.115 | 0.024 |
| H9b | Context: longer horizon | tent | 128 | 0.004 | 0.131 | 0.096 |
| H10a | System: logistic map | logistic | 64 | 0.004 | 0.053 | 0.022 |
| H10b | System: sinusoid family | sinusoid | 64 | 0.004 | 0.002 | 0.003 |
| H11a | Sensitivity: low-chaos regime | tent | 64 | 0.004 | 0.044 | 0.047 |
| H11b | Sensitivity: high-chaos regime | tent | 64 | 0.004 | 0.209 | 0.213 |
| H12 | Mitigation: control-label conditioning | tent | 64 | 0.004 | 0.008 | 0.007 |
| H13 | Mitigation: importance reweighting (post-hoc) | tent | 64 | 0.002 | 0.093 | 0.061 |

Table 1: **Tent-map hypothesis ladder (full numerics).** Each row is one ablation under a shared recovery interface. TV is computed from posterior-mixture pullbacks; "pt" reports TV for a point-estimator pullback when available.

$\times$**Hypothesis H1:** CM *is an artifact of a poorly specified recovery estimator.*

We hold the generator fixed and vary recovery estimation choices that could plausibly inject systematic bias: we swap the $\ell_2$ discrepancy inside the recovery likelihood for $\ell_1$ (H1a), and for sample-level rollout diagnostics we swap the point estimate extracted from the posterior between its mode and posterior mean (H1b/H1c). These changes do not remove the recovered-control shift: H1a yields $\text{TV}_{\text{model}} = 0.122$ (mixture), H1b yields $\text{TV}_{\text{model}} = 0.128$, and H1c yields $\text{TV}_{\text{model}} = 0.122$, with the data baseline remaining at $\text{TV}_{\text{data}} = 0.004$. We therefore reject H1 (Fig. 7).

$\times$**Hypotheses H2–H3:** CM *is driven by insufficient exposure to some controls or by under-training.*

We test whether CM disappears when we (i) rebalance control coverage explicitly by stratifying the training mixture over $r$ (H2), or (ii) vary training duration (H3a/H3b). Neither change restores recovered-control coverage under the same recovery interface: $\text{TV}_{\text{model}} = 0.126$ for H2, $\text{TV}_{\text{model}} = 0.129$ for short training (H3a), and $\text{TV}_{\text{model}} = 0.122$ for long training (H3b), with $\text{TV}_{\text{data}} = 0.004$ throughout. We therefore reject H2–H3.

$\times$**Hypothesis H4:** CM *is a capacity bottleneck.*

Increasing denoiser width (H4) does not remove CM under the same pipeline and recovery: $\text{TV}_{\text{model}} = 0.125$ (vs. $\text{TV}_{\text{data}} = 0.004$). While larger models often reduce observation-space error, these results suggest that higher capacity alone does not guarantee the *control* distribution implied by the fixed recovery procedure. We therefore reject H4 as a sufficient explanation.

$\times$**Hypothesis H5:** CM *is a locality artifact of the convolutional denoiser.*

We increase the convolution kernel size (H5) to change the inductive bias toward local structure. This does not eliminate misalignment: $\text{TV}_{\text{model}} = 0.127$. We therefore reject H5.

$\times$**Hypothesis H6:** CM *is diffusion-specific; other unconditional model families should avoid it.*

We replace the diffusion denoiser with alternative unconditional families under the same recovery interface: an MLP diffusion denoiser (H6a), an autoregressive Transformer (H6b), and a VAE baseline (H6c). None of these family changes reliably removes CM: $\text{TV}_{\text{model}} = 0.309$ (H6a), $\text{TV}_{\text{model}} = 0.261$ (H6b), and $\text{TV}_{\text{model}} = 0.253$ (H6c); the autoregressive baseline is particularly poor in this setting (Fig. 7). We therefore reject H6.

$\times$**Hypotheses H7–H9:** CM *is an artifact of sampler choice, objective parameterization, or observation window length.*

Changing the sampler (DDIM; H7), switching to an $x_0$-prediction parameterization (H8), or varying horizon length (H9a/H9b) does not restore recovered-control coverage under the same recovery: $\text{TV}_{\text{model}} = 0.126$ (H7), $\text{TV}_{\text{model}} = 0.126$ (H8), $\text{TV}_{\text{model}} = 0.115$ (H9a), and $\text{TV}_{\text{model}} = 0.131$ (H9b). We therefore reject H7–H9.

$\times$**Hypothesis H10:** CM *is idiosyncratic to the tent map.*

We repeat the same pipeline on related one-dimensional dynamical families. Misalignment persists in the logistic map (H10a; $\text{TV}_{\text{model}} = 0.053$), but is substantially reduced in the sinusoid family (H10b; $\text{TV}_{\text{model}} = 0.002$), where controls remain well-separated in observation space under the same recovery interface. This indicates that CM depends on the *control geometry induced by recovery* rather than on any single system. We therefore reject H10 as stated.

$\checkmark$**Hypothesis H11:** *Sensitivity amplifies* CM.

To isolate sensitivity within the tent family, we compare a low-chaos regime (H11a) to a high-chaos regime (H11b). Recovered-control misalignment is noticeably larger in the high-chaos setting: $\text{TV}_{\text{model}} = 0.044$ (H11a) versus $\text{TV}_{\text{model}} = 0.209$ (H11b). This matches the main-text mechanism: when the forward map becomes more sensitive, small structured deviations in observation space are more easily transported into a coherent recovered-control shift. We therefore support H11.

$\checkmark$**Hypothesis H12:** *When controls are known before training, conditioning restores coverage.*

Training the conditional model $p_\theta(x \mid r)$ and sampling $r \sim \pi$ at generation (H12) largely restores recovered-control alignment under the same recovery interface: $\text{TV}_{\text{model}} = 0.008$. This supports the main-text message that conditioning can bypass recovery-induced misalignment when the relevant control is available at training time.

$\times$**Hypothesis H13:** CM *is mainly a misallocation effect that can be repaired by importance reweighting.*

We test post-hoc importance reweighting of generated samples using the recovered-control posterior (H13). This reduces but does not eliminate misalignment: $\text{TV}_{\text{model}} = 0.093$ (with $\text{TV}_{\text{data}} = 0.002$ under the same sampling estimator). This matches the fact that reweighting can only change the

frequency of existing outputs, not the recovery ambiguity structure induced by off-manifold samples under a fixed recovery interface. We therefore reject H13 as a general remedy.

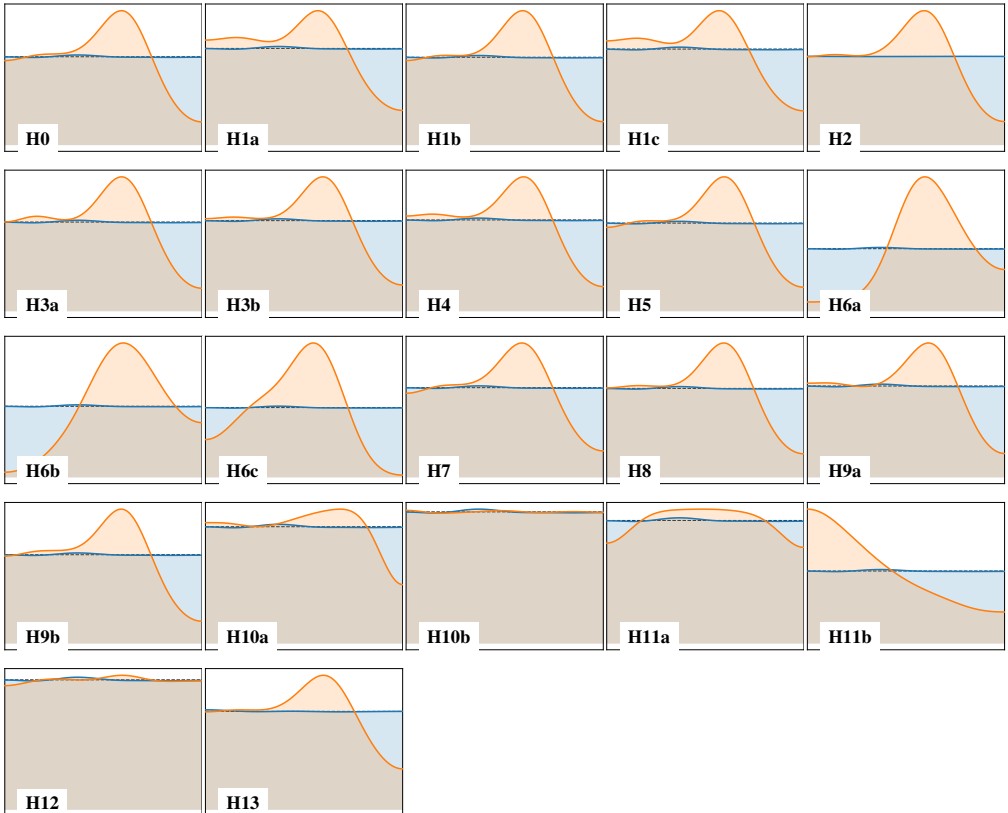

Figure 7: **Tent-map hypothesis ladder: recovered-control pullbacks.** Each mini is labeled by its hypothesis tag (see Tab. 1 for numerics).

# G  ADDITIONAL VISUALIZATIONS: RECONSTRUCTION DIAGNOSTICS

To help interpret CM beyond aggregate pullbacks, we include reconstruction diagnostics for representative trajectories. For a trajectory $x$, we compute a point estimate $\hat{r}(x)$ using the experiment's recovery estimator, and compare $x$ to the forward rollout $g(\hat{r}(x))$ under the same horizon. For ground-truth trajectories, this serves as a sanity check that recovery produces simulator-consistent reconstructions. For model trajectories, the same reconstruction can look plausible even when the recovered-control pullback is misaligned, because a good point estimate for individual samples does not guarantee that the induced mixture over controls matches the intended prior.

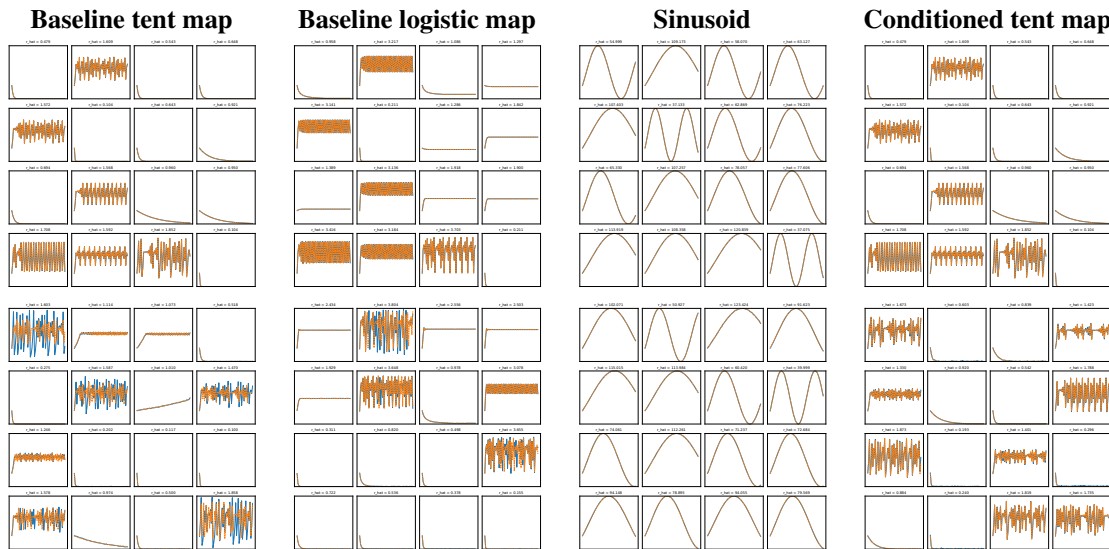

Figure 8: **Reconstruction diagnostics for synthetic benchmarks.** Top row: ground-truth trajectories and their reconstructions at $\hat{r}(x)$. Bottom row: model trajectories and their reconstructions at $\hat{r}(x)$. Even when sample-level reconstructions appear simulator-consistent, the recovered-control pullback can remain misaligned in sensitive regimes, while conditioning reduces CM.

# H AMBIGUITY BENCHMARK AND ITERATIVE PRIOR CORRECTION

The tent-map ladder above largely isolates *sensitivity*. This section isolates a complementary amplifier: *ambiguity* in the recovery interface. Here, many observations are comparably compatible with multiple distant controls, so even a small amount of off-manifold observation-space mass can produce a coherent recovered-control shift.

**Iterative prior correction (IPC).** IPC is a training-time resampling procedure: at iteration $k$, train a model on a resampled training mixture, estimate its recovered-control pullback $\bar{\pi}^{(k)}$, and then reweight the next training mixture by $\pi(r)/\bar{\pi}^{(k)}(r)$ evaluated at each training example's recovered control. In ambiguity-dominated settings, resampling can change how often certain *observations* appear, but it cannot remove the structural fact that many observations induce multi-modal recovery posteriors under the fixed recovery interface.

| IPC iter | $n_{\text{train}}$ | $n_{\text{eval}}$ | TV (data) | TV (model) |
|---|---|---|---|---|
| 0 | 60000 | 20000 | 0.022 | 0.119 |
| 1 | 60000 | 20000 | 0.022 | 0.084 |
| 2 | 60000 | 20000 | 0.022 | 0.069 |
| 3 | 60000 | 20000 | 0.022 | 0.062 |
| 4 | 60000 | 20000 | 0.022 | 0.059 |
| 5 | 60000 | 20000 | 0.022 | 0.057 |

Table 2: **IPC numerics for the ambiguity benchmark.** Per-iteration total variation of recovered-control pullbacks under the fixed recovery interface.

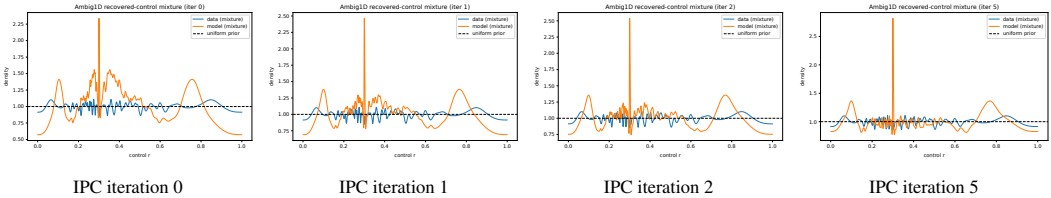

| IPC iteration 0 | IPC iteration 1 | IPC iteration 2 | IPC iteration 5 |

Figure 9: **IPC progression under ambiguity (folded-map benchmark).** Training-time resampling reduces recovered-control misalignment but can plateau above the data baseline, consistent with the fact that many observations remain multiply compatible with distant controls under the fixed recovery interface.

# I   DEPLOYED VIGNETTES: ADDITIONAL DETAILS

**Numerical summary.**

| Vignette | Setting | $N$ | Mean (data) | Mean (model) | TV (data vs model) |
|----------|---------|-----|-------------|--------------|--------------------|
| Maze2D (speed) | uniform_speed | 1000 | 3.201 | 5.716 | 0.923 |
| Double pendulum (energy) | energy-conserving simulator, position-only training | 2000 | 22.599 | 27.467 | 0.118 |
| Tacotron2 (speaking rate) | ASR-based recovery, paired WER filter | 1000 | 4.232 | 4.386 | 0.126 |

Table 3: **Numerical summaries for the deployed vignettes.** We report mean recovered controls and total-variation discrepancies under the shared recovery procedures used in the main text.

## I.1   MAZE2D: SETUP AND RECOVERY

**Setup and recovery.**   We use the D4RL Maze2D U-maze dataset (Fu et al., 2020). We treat trajectory speed as an implicit control and recover it from a plan as the mean path length per unit time,

$$v(x) = \frac{1}{(H-1)\Delta t} \sum_{t=0}^{H-2} \|q_{t+1} - q_t\|_2,$$

where $q_t$ is the position sequence and $\Delta t$ is estimated from the dataset. We construct a binned-uniform speed training mixture by filtering feasible expert trajectories and resampling by recovered speed. Concretely, we use horizon $H = 128$, $\Delta t = 0.0100648$, and a uniform target speed range $v \in [2.7, 3.7]$ (chosen between dataset quantiles 30 and 75). For the recovery grid we use 20 uniform bins over the target range, map bins linearly to $r \in [0, 1]$, and use the adaptive Gaussian recovery with $\sigma_r = 0.05$ (clipped to $[0.01, 0.2]$). When reporting feasibility-based diagnostics, we use an occupancy grid of resolution 200, dilate obstacles for 2 iterations, and treat a rollout as valid if its maze-support fraction exceeds 0.95.

**Iso-MSE jitter probe.**   Let $q_{0:H-1}$ be a valid trajectory segment and $q'_{0:H-1}$ a perturbed version with fixed endpoints ($q'_0 = q_0$ and $q'_{H-1} = q_{H-1}$). Define the displacement sequence $d_t := q'_t - q_t$ and its second difference

$$\Delta^2 d_t := d_{t+2} - 2d_{t+1} + d_t, \qquad t = 0, \ldots, H - 3.$$

Our iso-MSE construction scales perturbations so that the trajectory MSE budget

$$\text{MSE}(q', q) := \frac{1}{H} \sum_{t=0}^{H-1} \|d_t\|_2^2$$

is fixed, and then varies only the temporal roughness of $d_{0:H-1}$. We summarize roughness by the dimensionless normalized jitter score

$$J(q', q) := \sqrt{\frac{\frac{1}{H-2} \sum_{t=0}^{H-3} \|\Delta^2 d_t\|_2^2}{\text{MSE}(q', q)}}.$$

## I.2   DOUBLE PENDULUM: SETUP AND RECOVERY

**Setup.**   We train a diffusion model on position-only trajectories and recover energy using finite differences and the standard energy functional. Model generations can violate energy conservation, so the recovered energy can drift substantially over time, and there is no unique invariant energy value for recovery to target. This contributes an ambiguity-like component in the pendulum vignette in addition to recovery sensitivity. Our diffusion model uses the JannerUNet1d backbone with base width 32, kernel size 3, channel multipliers $(1, 2, 2, 2)$, and a 256-step diffusion schedule; we train for 50 epochs with batch size 64 and learning rate $10^{-3}$ on position-only trajectories, with angles normalized to a fixed range. Unless stated otherwise, quantitative summaries use $N = 2000$ samples under the same recovery.

### I.3 TACOTRON2: SPEAKING-RATE RECOVERY

**Recovery.** We recover speaking rate by syllables-per-second after filtering based on ASR, as described in the main text (Sec. 2.1). We use the same recovery pipeline for data and model generations. Concretely, we use the LJSpeech dataset and synthesize speech with a pretrained Tacotron2 system and neural vocoder. Speaking rate is recovered using a pretrained wav2vec2 ASR model at 16kHz, over utterances with duration in $[0.3, 20]$ seconds and $\geq 2$ words. We keep 1000 utterances and apply a paired word error rate filter; in the reported run, 757 paired utterances remain after filtering.

