# OpenReview forum: "When does Observational Data Teach Latent Dynamics? Understanding Control Misalignment with Synthetic Tasks"
_ICLR.cc/2026/Workshop/Sci4DL — Sci4DL 2026_

### Official Review · Reviewer_TU6E · 2026-02-16

**Fit:** 3
**Significance:** 3
**Confidence:** 2

**Summary:**

This paper presents unconditional generative models that align with the observation marginal while distorting the distribution of latent controls. The authors demonstrate this concept (CM) in applications such as Maze2D speed, pendulum energy, and Tacotron2 speaking rate. They identify the mechanism using synthetic sensitivity benchmarks and show that recovery procedures can amplify minor observation errors into significant control space shifts, with conditioning largely addressing this issue.

**Strengths:**

- Strong mechanistic analysis connecting generative modeling to inverse-problem theory.

- Real-world vignettes + mitigation analysis (conditioning vs. reweighting).

**Suggestions:**

- Relies heavily on a specific recovery interface; conclusions depend on recovery design.

- Primarily 1D synthetic systems for theory validation.

---

### Official Review · Reviewer_mNpC · 2026-02-26

**Fit:** 3
**Significance:** 2
**Confidence:** 2

**Summary:**

The paper investigates Control Misalignment in deep generative models trained unconditionally on observational data. The authors demonstrate that even when models generate high quality samples they often fail to capture the underlying distribution of latent control parameters like speed or energy. They show this phenomenon across real world tasks including Maze2D and physical simulations  and then use minimal synthetic tasks to prove that recovery sensitivity and ambiguity are the root causes.

**Strengths:**

* The identification and formalization of Control Misalignment across three diverse domains provides a strong practical motivation for the work
* The use of minimal synthetic benchmarks like the tent map and folded map to isolate sensitivity and ambiguity is elegant and highly effective for explaining the root causes
* The hypothesis formulation in the appendix is extremely thorough and systematically rules out alternative explanations like capacity bottlenecks or poor hyperparameter tuning

**Suggestions:**

Weakness:
* The mitigations discussed starting at line 186 rely heavily on conditional modeling which requires access to control labels during training and somewhat bypasses the core challenge of unconditional alignment
* The formal definition of Control Misalignment on line 80 depends entirely on the chosen recovery procedure which means recovery artifacts could still conflate the metric despite the authors shared interface approach
* The theoretical bound introduced on line 84 linking total variation to the Wasserstein distance relies on a Lipschitz constant L that is practically unknowable for the complex real world vignettes presented like Tacotron2

Suggestions/Questions:
* Could you expand on potential mitigations for Control Misalignment that do not require explicit control labels during training? The conditional modeling mitigation is effective but somewhat bypasses the core challenge of unconditional alignment.
* The Iterative Prior Correction method requires retraining the model on reweighted mixtures. It would be beneficial to add a brief discussion on the computational overhead and scalability of this mitigation approach for larger real-world models.
* You demonstrate that high capacity alone does not guarantee the control distribution implied by the fixed recovery procedure. Are there any specific architectural inductive biases beyond capacity and convolution kernel sizes that you hypothesize might naturally resist Control Misalignment?

---

### Official Review · Reviewer_4B7y · 2026-02-27

**Fit:** 2
**Significance:** 2
**Confidence:** 2

**Summary:**

The paper refers to systems that use unconditional generators as surrogates for control systems. The authors specifically study when unconditional generative modelling of an observation marginal $q(x)$ distribution fails to match the intended latent control parameter distribution ($r$). The authors refer to this failure as **control misalignment** and study when even even if a model marginal $q_{\theta}(x)$ matches $q(x)=\int p(x\mid r)\pi(r)dr$, the distribution over $r$ recovered from the generated samples can differ substantially from the intended prior $\pi(r)$.

The authors consider a recovery procedure that maps from an observation $x$ to a distribution over latent controls $r$, $\rho(r\mid x)$, and define the pullback/recovered control marginal as
$\bar\pi_S(r)= E_{x\sim S}[\rho(r\mid x)]$ for sources $S$. They quantify  misalignment in terms of the total variation distance between $\bar\pi_{q_\theta}$ and $\pi$, and in particular they posit that there is control misalignment when the total variation distance between
the recovered-control distribution (obtained through the recovery procedure $\rho$) and the intended control prior (baseline recovery error) is much smaller than the total variation between the recovered-control distribution from model samples and the intended control prior (recovered control misalignment to prior).


The paper tests this metric on synthetic tasks with a know mapping form latent controls to observations, a  Maze2D trajectory planning setting, a double pendulum simulation, and Tacotron2 speech synthesis.
 Moreover they provide a mechanistic explanation framed as density transport through recovery for this failure that posits that structured observation-space errors that are low-cost under the marginal objective may become large recovered-control shifts, when the control-recovery is sensitive or ambiguous/non-identifiable .
They validate this proposed mechanism on synthetic benchmarks and form a thorough hypothesis ladder to rule out simpler explanations.

The authors further test for ways to mitigate this issue by increasing model capacity, by considering an iterative prior correction, and through conditional modeling, with the latter appearing to be the most effective mitigation strategy.

**Strengths:**

- Extensive empirical evidence across different domains where generative modelling is relevant (motion planning, physical simulation, and speech synthesis)
- Empirical evidence that control misalignment can be large even when samples look plausible, and that common changes (capacity, model family, sampling method) do not necessarily remove it, while conditioning on controls $r$ can.
- The transport identity $\tau_\theta(r)=\int \Delta_\theta(x)\rho(r\mid x)\text{d}x$ provides a concrete mechanism linking small observation-space modeling artifacts to shifts in recovered controls.
- I particularly like the hypothesis ladder ablation table, which shows that control misalignment is prevalent across multiple architectures/samplers, but falls near baseline, when training is conditioned on the control label.

**Suggestions:**

- So as I understand control misalignment can happen either due to misalignment of the generator with the real data distribution, or because the control-recovery step is overly sensitive, so that small samples in the observation space cause big changes in the recovered control distribution. The proposed metric precisely considers both potential failure modes, however it would be insightful if in the numerical experiment you would test which failure mode is causing the observed control misalignment. Can you separately quantify each contributing factors in each case to understand whether indeed it is the sensitivity of the control-recovery that causes the misalignment?

- Can you suggest a threshold/criterion or statistical test on how exactly to decide when there is control misalignment or not.
- In Table 3, Maze2D speed has very large total variation misalignment. Can you rule out the possibility that the generator is matching $q(x)$ poorly in a way that would be visible with observation space metrics, i.e., quantify the Wasserstein distance or another observation-space distance and relate it to control misalignment?
- Can you make a systematic sensitivity analysis over grid size and bandwidth hyper parameters of the recovery procedure to quantify whether the control misalignment results change?

---

### Meta-Review · Area_Chair_AP7j · 2026-03-01

**Recommendation:** Accept

**Metareview:**

Recommending acceptance. Suggest the authors incorporate the suggestions from the reviewers.

---

### Decision · Program_Chairs · 2026-03-02

Accept